# Genomic data from NSCLC tumors reveals correlation between SHP-2 activity and PD-L1 expression and suggests synergy in combining SHP-2 and PD-1/PD-L1 inhibitors

Keller J. Toral, Mark A. Wuenschel, Esther P. Black[ID]*

Department of Pharmaceutical Sciences and Markey Cancer Center, College of Pharmacy, University of Kentucky, Lexington, Kentucky, United States of America

* penni.black@uky.edu

**Data Availability Statement:** All data used in this study were accessed from 3 published studies and publicly-available RNA sequencing data from NCI-

## Abstract

The identification of novel therapies, new strategies for combination of therapies, and repurposing of drugs approved for other indications are all important for continued progress in the fight against lung cancers. Antibodies that target immune checkpoints can unmask an immunologically hot tumor from the immune system of a patient. However, despite accounts of significant tumor regression resulting from these medications, most patients do not respond. In this study, we sought to use protein expression and RNA sequencing data from The Cancer Genome Atlas and two smaller studies deposited onto the Gene Expression Omnibus (GEO) to advance our hypothesis that inhibition of SHP-2, a tyrosine phosphatase, will improve the activity of immune checkpoint inhibitors (ICI) that target PD-1 or PD-L1 in lung cancers. We first collected protein expression data from The Cancer Proteome Atlas (TCPA) to study the association of SHP-2 and PD-L1 expression in lung adenocarcinomas. RNA sequencing data was collected from the same subjects through the NCI Genetic Data Commons and evaluated for expression of the PTPN11 (SHP-2) and CD274 (PD-L1) genes. We then analyzed RNA sequencing data from a series of melanoma patients who were either treatment naïve or resistant to ICI therapy. PTPN11 and CD274 expression was compared between groups. Finally, we analyzed gene expression and drug response data collected from 21 non-small cell lung cancer (NSCLC) patients for PTPN11 and CD274 expression. From the three studies, we hypothesize that the activity of SHP-2, rather than the expression, likely controls the expression of PD-L1 as only a weak relationship between PTPN11 and CD274 expression in either lung adenocarcinomas or melanomas was observed. Lastly, the expression of CD274, not PTPN11, correlates with response to ICI in NSCLC.

## Introduction

Lung cancers remain a leading cause of cancer morbidity and mortality worldwide despite increased efforts toward drug discovery and implementation of personalized medicine

GDC. Specifically, 1) for NCI-GDC, we used the "TCGA-LUAD" data set (similarly, the TCPA used the TCGA-LUAD-L4 subset); 2) for PMID: 30388455, GEO accession# GSE115978; 3) for PMID: 31959763, GEO accession# GSE136961; and 4) for PMID: 32015526, we used the full dataset.

**Funding:** The authors received no specific funding for this study.

**Competing interests:** The authors have declared that no competing interests exist.

approaches [1]. Perhaps the most significant advance in therapy for many cancer types was the entry of immune checkpoint inhibitors (ICI) as a standard of care therapy for melanomas in 2014 [2]. For non-small cell lung cancers (NSCLC), specifically those without targetable mutations in the epidermal growth factor receptor (EGFR) or anaplastic lymphoma kinase (ALK), immune checkpoint inhibitors, specifically the antibodies that target programmed cell death 1 (PD-1) or programmed death ligand 1 (PD-L1), have revolutionized cancer therapy even though response rates are relatively low [3]. Both pembrolizumab and atezolizumab are approved ICI for frontline lung adenocarcinoma therapy for patients with high levels of PD-L1 expression on tumor cells [4]. Durvalumab, an anti-PD-1 agent, is approved as maintenance therapy [5]. Decisions to implement ICI therapy is often dependent on the PD-L1 tumor proportion score using evidence from the KEYNOTE-024 and -042 trials [6, 7]. Importantly, PD-L1 expression may not be the optimal biomarker of response as suggested in pivotal clinical studies (e.g. KEYNOTE and OAK), but it is clear that patients with high levels of tumoral PD-L1 are likely to experience a robust response to checkpoint inhibition [8]. While many research groups have searched for improved biomarkers of response for checkpoint inhibitors, others have focused on identification of therapies that might be combined with ICI to improve patient outcomes. The work presented herein falls into the latter category [9].

Our group found that inhibition of the tyrosine phosphatase, SHP-2, increased gene and cell surface protein expression of PD-L1 in KRAS-active NSCLC cell lines (manuscript submitted). PD-L1 is normally expressed on the surface of antigen presenting cells while PD-1 is expressed on T cells. It is the abnormal expression of PD-L1 on tumor cells, and the subsequent engagement with PD-1 on T cells, that causes tumors to be masked from an immune response [10]. Inhibiting this interaction with antibodies against either PD-1 or PD-L1 can release a potent immune response toward the tumor.

We hypothesized that because SHP-2 provides some control of expression of PD-L1 on NSCLC cells that inhibition of SHP-2 would increase PD-L1 expression and synergize with ICI therapy. Supportive of our hypothesis is recently published data by Chen and colleagues showed in a NSCLC model system that combined SHP2 and PD-L1 inhibition, with accompanying radiation, can overcome resistance to PD-1 inhibitors [11]. Other groups have suggested that SHP-2 activity may be more important in T cells, that infiltrate the tumor, to carry out signaling events downstream of PD-1 stimulation [12]. Uncovering the precise mechanism of SHP-2 action on PD-L1 expression consumes many research groups, the model systems are expensive, and experimental time is long to get a drug to the clinic. In this study, we chose to go straight to real world data to determine whether SHP-2 activity is related to PD-L1 expression and thereby focus our research efforts.

We took advantage of three publicly-available data sets to assess whether moving forward with wet lab experimentation to determine if exploring the combination of ICI and SHP-2 inhibition is warranted. First, The Cancer Genome Atlas, now known as the NCI Genetic Data Portal (NCI-GDC), holds well-annotated expression and functional proteomic data (The Cancer Proteome Atlas (TCPA)) for patient tumors. However, most samples were collected prior to FDA approvals for ICI therapy, so no response data for ICI treatment is available (https://portal.gdc.cancer.gov/projects/TCGA-LUAD). Unfortunately, larger, industry-sponsored trials are still open (e.g. KEYNOTE and OAK), and full genomic and patient response datasets are not yet published. Therefore, in order to link expression of SHP-2 and PD-L1 with response to ICI, we uncovered two small studies: one in NSCLC and one in melanoma patients [13, 14]. Using real world data from the three studies identified, we believe that inhibition of SHP-2 activity is likely to improve response to PD-L1/PD-1 inhibitors and justifies continue wet-lab characterization of the mechanism(s) of activity.

## Methods and results

### Phosphorylated SHP-2 significantly correlates with the loss of PD-L1 expression in lung adenocarcinomas

First, using TCPA (https://tcpaportal.org/tcpa/index.html), a functional proteomics database which contains reverse phase protein array (RPPA) data from a wide variety of clinical tumor samples, we identified a lung adenocarcinoma (TCGA-LUAD-L4) dataset containing RPPA data from 362 individual patient samples. These data contain quantitative protein expression levels of 237 unique proteins for each subject.

From the TCPA data, SHP-2_ pY542, the phosphorylated and active form of SHP-2, and PD-L1 were compared from 362 patient tumors for relative protein expression levels using a two-tailed, non-parametric Spearman correlation analysis with 95% confidence intervals. The analysis revealed that levels of SHP-2_pY542 negatively correlate with PD-L1 expression (r = -0.157, p-value = 0.0028$^{**}$) in these subjects, suggesting that inhibition of SHP-2 activity may increase PD-L1 protein expression (Figs 1A and 2A). Data capture and analysis was automated, and the annotated code in Python is linked here: (https://github.com/mwu228/Summer2021/blob/main/Correlate%20proteins%20of%20interest.ipynb).

The data contained in the TCPA contained only expression levels of the active, phosphorylated SHP-2, so we were unable to compare interactions with the unphosphorylated form. Thus, we looked to proteins in the TCPA dataset known to be in signaling cascades controlled by SHP-2 activity as internal controls, specifically Src, STAT3, and MAPK (need references here). We conducted the same correlation analysis between SHP-2_pY542 and either Src_pY527, Src_pY416, STAT3_pY705, or MAPK_pT202Y204. We found that the levels of active SHP-2 maintain strong (r > 0.4) and statistically significant (p < $1x10^{-15}$) positive correlations with each of these four proteins, providing additional support that the Y542 phosphorylation of SHP-2 correlation with PD-L1 protein expression is a meaningful interaction (S1 Fig).

Next, to better understand the relationship between expression of SHP-2 (PTPN11) and PD-L1 (CD274) mRNA in these patient tumors, we acquired corresponding RNA-sequencing data from TCGA, now known as the NCI-GDC (https://portal.gdc.cancer.gov). In this database, the TCGA-LUAD dataset contained 585 tumor samples, 223 more than the TCPA data. To look at only RNA-sequencing data that matched the previously-queried RPPA data, the 362 patient identifiers provided by the TCPA database were used to identify the corresponding RNA-sequencing data deposited into the GDC. We utilized fragments per kilobase-upper quartile (FPKM-UQ) values. The FKPM-UQ values for the genes PTPN11 and CD274 for each tumor were subjected to the same Spearman correlation analysis as previously described. Data capture and analysis was automated, and the annotated code in Python is linked here: (https://github.com/mwu228/Summer2021/blob/main/RNAseq%20FPKM%20correlation%20and%20pvalue.ipynb).

Interestingly, this analysis revealed no significant correlation (p-value = 0.3488) between PTPN11 and CD274 mRNA expression levels (Fig 1B). Following this observation, we wanted to know if any relationship between PTPN11 and CD274 expression was found using the entire TCGA-LUAD RNA-sequencing dataset (n = 585). We found a slight positive correlation existed (r = 0.095, p-value = 0.0211$^*$) between PTPN11 and CD274 mRNA levels (Fig 1C). Because we are most interested in the role of SHP-2 in KRAS-active LUAD, we sub-grouped tumors with variants of KRAS known to be active from this dataset. No significant relationship was found in KRAS-active lung adenocarcinomas between PTPN11 and CD274 levels (S2 Fig).

Extending our observations from the aforementioned data that suggest a relationship between SHP-2 activity and PD-L1 expression, we identified another data warehouse

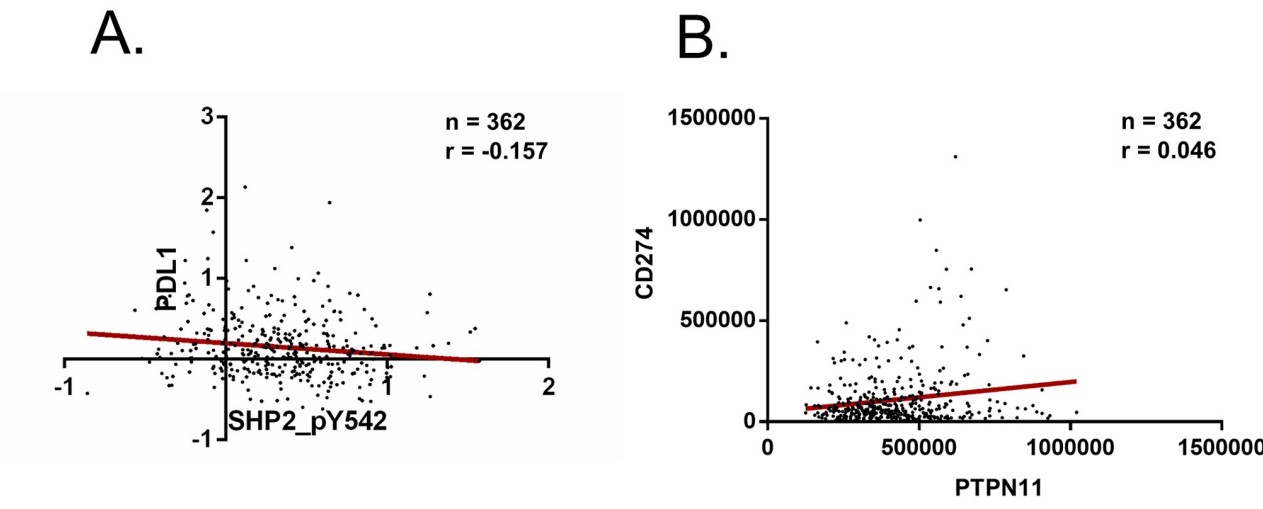

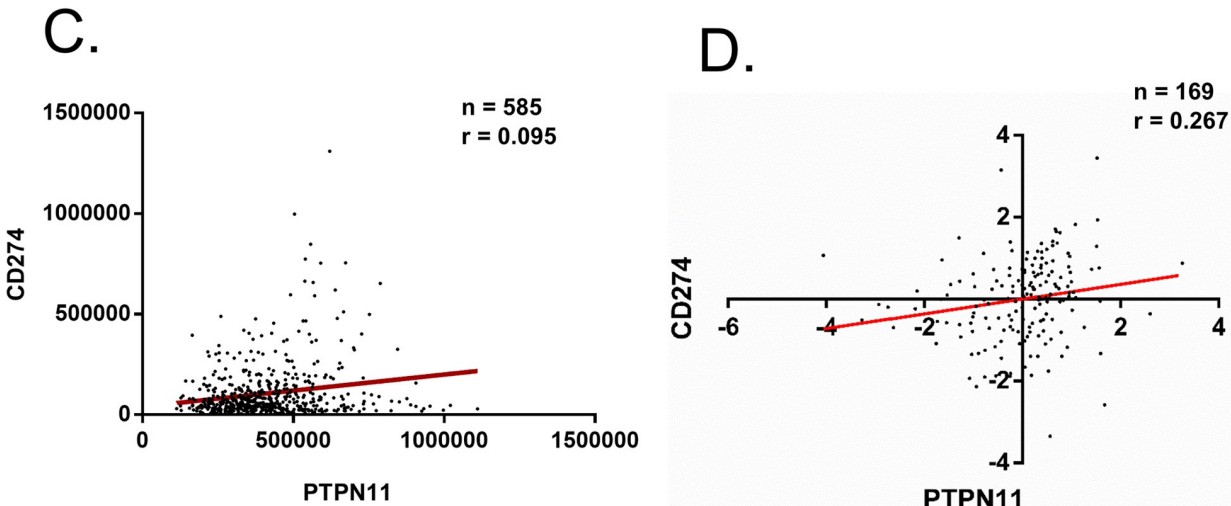

**Fig 1. SHP-2 activity and expression correlates with expression of PD-L1 in NSCLC adenocarcinomas. A**. Two-tailed non-parametric Spearman correlation analysis of RPPA protein expression data for Y542 phosphorylated SHP-2 and PD-L1 from 362 adenocarcinomas taken from The Cancer Proteome Atlas (TCPA: https://gdc.cancer.gov/about-data/publications/pancanatlas) LUAD-L4 dataset. **B**. Two-tailed non-parametric Spearman correlation analysis of bulk RNA-seq FPKM-UQ values taken from TCGA (GDC) for the 362 patients that had corresponding RPPA protein expression data from TCPA. **C**. Two-tailed non-parametric Spearman correlation analysis of bulk RNA-seq FPKM-UQ values taken from TCGA (GDC) for all 585 patients in the TCGA-LUAD dataset. **D**. Two-tailed non-parametric Spearman correlation analysis of mRNA z-scores taken from cBioPortal (PMID:32015526) for 169 lung adenocarcinoma tumors. The red line in each panel represents a linear regression line of best fit.

(cBioPortal: cbioportal.org) that contains gene expression data from clinical cancer studies. Specifically, we located a study sought to characterize the genomic landscape of lung adenocarcinomas in East Asians [15]. This study contains RNAseq data for 169 patients, from which we conducted a two-tailed, non-parametric Spearman correlation analysis with 95% confidence intervals between PTPN11 and CD274 mRNA levels (normalization method: z-score). The analysis revealed a positive ($r = 0.267$) and significant (p-value = 0.0005***) correlation

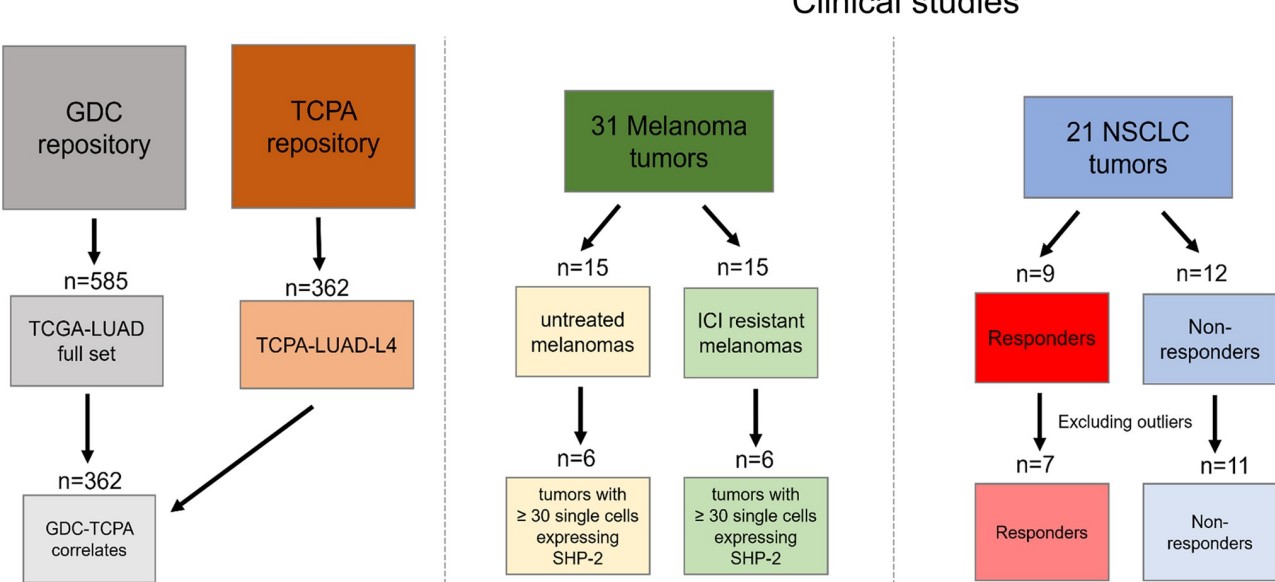

**Fig 2. Workflow scheme for evaluation of SHP2 and PD-L1 relationships.** Reverse-phase protein array (RPPA) was collected from the TCPA data repository (https://gdc.cancer.gov/about-data/publications/pancanatlas) for 362 total patients labeled as the TCPA-LUAD-L4 data set. RNAseq data was collected from GDC for the full TCGA-LUAD dataset (n = 585). These data were parsed to include only patients for which there was matching RPPA data on TCPA (n = 362). Single-cell RNAseq reads for 31 melanoma tumors were collected and separated into two groups based on ICI treatment status. Only single-cell reads for 'malignant melanoma cells' were retained for analysis. Tumors which had ≥ 30 unique malignant cells with non-zero PTPN11 values were included in the analysis. NSCLC tumors (n = 21) with sequence data were first separated into two groups based on response to ICI treatment. Average TPM values were calculated for PTPN11 and CD274, and tumors that had PTPN11 TPM value >2 standard deviations from the mean were excluded from the analysis.

between PTPN11 and CD274 mRNA, again suggesting that SHP-2 and PD-L1 protein are co-expressed in LUAD tumors (Fig 1D). Together, these TCPA and RNA seq data suggest that SHP-2 and PD-L1 protein are co-expressed in LUAD tumor tissue and that activation of SHP-2, not simply expression, may control levels of PD-L1. However, without knowing the expression levels of inactive SHP-2, we cannot state with certainty that SHP-2 activity is the primary role by which SHP-2 regulates PD-L1 expression.

## PTPN11 mRNA expression weakly associates with reduced CD274 mRNA expression in melanoma tumors regardless of ICI exposure

Having established a connection between tumoral SHP-2 activity and PD-L1 expression, but not corresponding gene expression levels, in lung adenocarcinomas, we sought to understand whether PTPN11 and CD274 expression levels associate with response of patient tumors treated with ICIs. A study was identified that analyzed single-cell RNA-sequencing (scRNA-seq) data from 31 melanoma tumors that were either not treated with ICIs or became resistant to ICIs following treatment. Importantly, the authors of the study were interested in characteristics of the melanoma cells that lead to immune evasion [13]. We used the R-studio Bioconductor GEOquery package to download the raw scRNA-seq transcript-per-million (TPM) values, cell counts, and annotations from this study (GSE115978). TPM values were calculated as described in Jerby-Arnon L., et al. [13]. We sought to answer two main questions using these data: 1) does PTPN11 mRNA expression correlate with CD274 mRNA levels and 2) does

PTPN11 expression correlate with poor response to PD-1 inhibition? The scheme for the analysis workflow is found in Fig 2B.

To address the first question, we identified the tumors which were not treated with ICI (n = 15). For each of these samples, we established that scRNA-seq reads were available for several cell types, including immune cell types and malignant cells. Cell types were detected by fluorescence activated cell sorting using cell-type specific proteins. Because we are only interested in associations between PTPN11 and CD274 in tumor cells, we selected only single cells determined to be malignant melanoma cells. Of the 15 untreated tumors, the analysis was narrowed to include patient tumors that have at least 30 unique malignant cells (n = 6) resulting in an average of 108 (range, 91–487) single-cells per tumor. To understand the proportion of single cells in an individual tumor that expressed PTPN11, the percentage of cells with non-zero TPM scores for PTPN11 for each tumor was calculated (Table 1).

This processing uncovered that six untreated tumors (Mel71, Mel79, Mel103, Mel80, Mel81, Mel89) demonstrated ≥50% of single malignant cells (mean = 69%; range, 50–83%) expressed PTPN11. The TPM values for PTPN11 and CD274 for all single malignant cells in these six tumors were then assessed together, resulting in mean/standard deviation TPM values for PTPN11(1.40, 0.22) and CD274 (0.11, 0.06). We observed a similar trend similar to that of the TCPA/NCI-GDC analysis above that elevated expression of PTPN11 associated with lower expression of CD274 in treatment naïve tumors.

Finally for this dataset, we wanted to understand the relationship of PTPN11 and CD274 expression and response to therapy. We used the patient tumors which acquired resistance to ICI therapy (n = 15) to ask whether the relative levels of PTPN11 and CD274 levels were different than the treatment-naïve tumors. Again, the data were processed to include only tumors

**Table 1.** A. PTPN11 mRNA expression weakly associated with reduced CD274 mRNA expression in melanoma tumors regardless of ICI exposure. B. PTPN11 mRNA expression weakly associated with reduced CD274 mRNA expression in melanoma tumors regardless of ICI exposure.

**A**

| Treatment | Tumor | % cells expressing PTPN11 | Tumoral PTPN11 TPM | Tumoral CD274 TPM | Average PTPN11 TPM | Average CD274 TPM |
|---|---|---|---|---|---|---|
| None | Mel80 | 80.4 | 1.35 | 0.07 | 1.40 | 0.14 |
| None | Mel81 | 81.7 | 1.57 | 0.02 | | |
| None | Mel89 | 83.0 | 1.78 | 0.18 | | |
| None | Mel71 | 56.5 | 1.24 | 0.07 | | |
| None | Mel79 | 58.8 | 1.40 | 0.18 | | |
| None | Mel103 | 50.4 | 1.08 | 0.12 | | |

**B**

| Treatment | Tumor | % cells expressing PTPN11 | Tumoral PTPN11 TPM | Tumoral CD274 TPM | Average PTPN11 TPM | Average CD274 TPM |
|---|---|---|---|---|---|---|
| Ipilimumab+nivolumab | Mel78 | 75.8 | 1.49 | 0.15 | 1.29 | 0.09 |
| Ipilimumab+pembrolizumab | Mel110 | 82.1 | 1.39 | 0.05 | | |
| Tremlimumab | Mel88 | 66.9 | 1.16 | 0.17 | | |
| Ipilimumab | Mel98 | 61.8 | 1.19 | 0.14 | | |
| Ipilimumab+nivolumab | Mel102 | 56.8 | 1.20 | 0.03 | | |
| Ipilimumab+pembrolizumab +nivolumab | Mel194 | 59.4 | 1.29 | 0.03 | | |

**A**. Six tumors had ≥ 30 unique malignant cells that had non-zero PTPN11 TPM values and are ICI therapy naïve [13]. Shown are the percentage of single cells that expressed PTPN11, the combined TPM values for all the single cells within each individual tumor, and the average TPM values for all six untreated tumors combined. **B**. The same details as **(A)** but for tumors that did not respond to ICI therapy. There was no significant difference in average PTPN11 and CD274 TPM values between treatment naïve and treatment resistant groups.

with ≥ 30 unique malignant cells (n = 6) resulting in an average of 79 single cells (range, 96–169) per tumor. We applied the methods used above to calculate the proportion of single cells expressing PTPN11 for each tumor, and the average TPM values for PTPN11 and CD274 when the single cells of all six tumors were evaluated together. We found that these six ICI resistant tumors (Mel78, Mel88, Mel98, Mel102, 196 Mel110, Mel94) again showed ≥50% single malignant cells (mean = 67%; range, 57–82%) expressed PTPN11, and the combined mean/standard deviation TPM values for PTPN11 (1.29, 0.12) and CD274 (0.09, 0.06). Here, similar expression patterns of PTPN11 and CD274 were observed compared with treatment-naive tumors, and again CD274 levels remain low when PTPN11 is expressed. Importantly, we were unable to observe any relationship between PTPN11 or CD274 expression and acquired resistance to ICI in this dataset.

## CD274, but not PTPN11, mRNA expression associated with response to ICI in NSCLC tumors

Using the data from the third study, we asked whether expression of PTPN11 and CD274 mRNA associates with response to ICI therapy in NSCLC. The investigators in this report aimed to find immune signatures predictive of response to anti- PD-1 inhibitors in NSCLC. The dataset contained bulk tumor RNA-sequencing data and clinical response data for 21 NSCLC subjects treated with single agent anti-PD-1 therapy [14]. As before, the R-studio Bioconductor GEOquery package was used to capture raw RNA-sequencing TPM values from this study (GSE136961). Patients demonstrating progression of disease or stable disease that lasted less than 24 weeks were deemed by the authors to have no durable clinical benefit (DCB) to anti-PD-1 therapy. Patients showing partial or complete response by Response Evaluation Criteria in Solid Tumor (RECIST) v1.1 or stable disease for more than 24 weeks were defined as receiving DCB. The analysis of these data followed a workflow scheme like that in Fig 2C.

Of the 21 NSCLC patients in this study, nine demonstrated a DCB to ICI therapy and twelve showed no DCB. We separated the data by DCB status and then averaged all TPM values for PTPN11 and CD274 for each patient tumor to generate one average TPM score for each group. An outlier analysis was performed on the PTPN11 TPM values for both responders and non-responders (S3 Fig), resulting in final groups of 7 responders (n = 7) and 11 non-responders (n = 11). Our analysis revealed no significant difference in the expression of PTPN11 mRNA between subjects with DCB from those that did not respond to anti-PD-1 therapy (Fig 3). Specifically, the mean/standard deviation PTPN11 TPM scores were (576.15, 281.78) and (487.73, 361.24) for responders and non-responders, respectively. Importantly, the mean expression of CD274 mRNA was nearly 3-fold higher in patients who responded to therapy (151.16, 198.33) compared to those who did not (61.96, 64.54). We note the standard deviation was large for the last two groups assessed. Together, these data showed that PTPN11 expression does not associate with CD274 expression or response therapy in NSCLC patients. These findings are consistent with the results from the first study that suggested that SHP-2 activity, not expression, correlates with PD-L1 expression. In contrast, these data demonstrated a positive relationship between PD-L1 expression and response to ICI which was not observed in the melanoma study.

## Discussion

In this study, we applied information obtained from publicly-available protein and gene expression datasets to gain further insight into our overarching research question: does SHP-2 activity or expression influence PD-L1 mRNA and protein levels and subsequent response to

A.

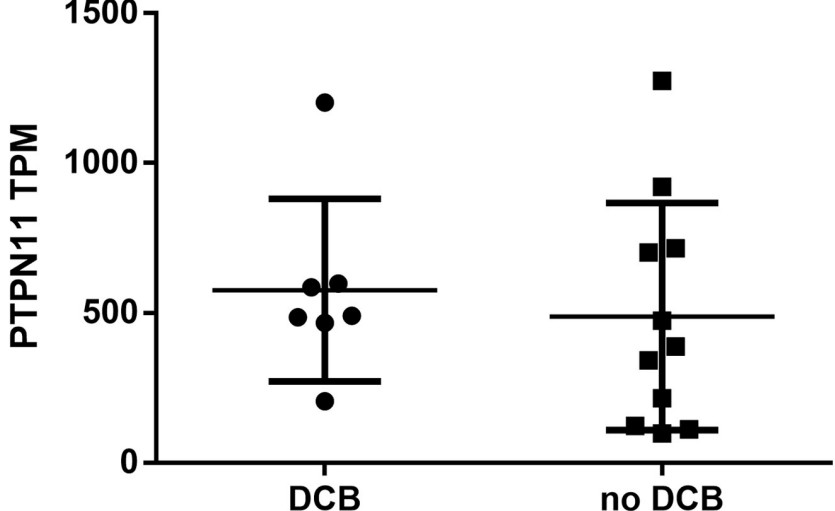

B.

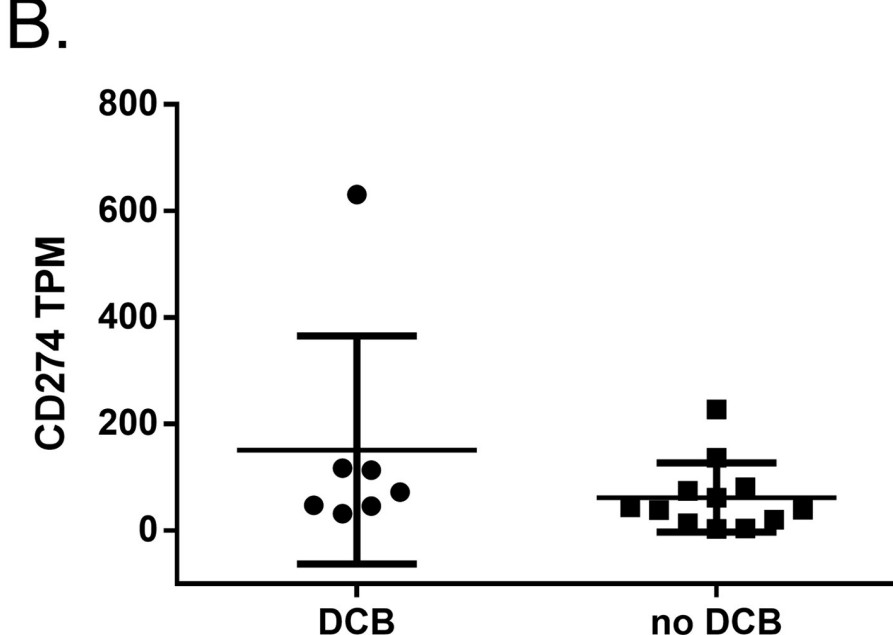

**Fig 3. CD274, but not PTPN11, mRNA expression is associated with response to ICI in NSCLC tumors. A.**
PTPN11 TPM values for patients who did or did not demonstrate a durable clinical benefit (DCB) from ICI therapy, as
determined by RECIST criteria [14]. There was no significant difference between groups, as measured by a student's t-
test. **B**. TPM values for CD274 in patients who did or did not demonstrate a durable clinical benefit (DCB) from ICI
therapy, as determined by RECIST criteria. There was no significant difference between groups, as measured by a
student's t-test.

anti-PD-1 or PD-L1 therapies in NSCLC? We used this approach because we believe that the utilization of real-world datasets can inform and direct wet-lab experimentation. The design and execution of pre-clinical and clinical studies is expensive, time-consuming, and labor-intensive. Here, we present a quick and efficient process that, when combined with bench-side techniques, can offer substantial insight into the clinical translatability of commonly-used, highly-controlled model systems designed for drug discovery applications. Through the analysis of two major cancer data repositories and two smaller clinical studies, we were able to take further steps towards establishing a connection between the activity of SHP-2 and PD-L1 expression in human tumors without carrying out a study *de novo*.

Of the datasets chosen for this study, the most statistically-powerful and revealing information arose from the composite analyses of the TCPA and GDC data repositories. Using genomic and protein information from a large cohort of NSCLC patients, our most important observation was the strong negative correlation (r = -0.157, p-value = 0.0028\*\*) between the active, tyrosyl-phosphorylated form of SHP-2 and PD-L1 protein expression (Fig 1A). A limitation of these data was that the RPPA data did not include expression levels of the unphosphorylated and inactive form of SHP-2 which would have been a useful control as informed by our wet-lab studies. However, to address this limitation, we investigated the relationship of SHP-2 at Y542 with known targets. We found strong positive correlations in the expression levels of SHP2_pY542 and three proteins (Src, MAPK, STAT3) whose activity is dependent on phosphorylation status and known to be regulated by SHP-2 activity [16–18]. While this is not a perfect control to confirm that the activity of SHP-2 predominates total SHP-2 expression as they relate to PD-L1 expression, it does provide additional evidence that quantifies association of SHP2_pY542 with known substrates.

The conformational changes induced by phosphorylation of SHP-2 could alter protein-protein interactions among signaling components and intracellular signaling cascades that impact PD-L1 expression [19]. Following from that hypothesis, we observed no statistically significant correlation between the levels of SHP-2 and PD-L1 mRNA in the patients in the NCI-GDC dataset that were initially studied in the TCPA dataset, again highlighting the potential importance of molecular interactions of SHP-2 dependent on its activated state. Interestingly, when we conducted the same analysis on the entire cohort of LUAD patients in the NCI-GDC repository, a weak, but inverse correlation, (r = 0.095, p-value = 0.0211\*) between PTPN11 and CD274 mRNA was observed, suggesting that SHP-2 and PD-L1 are co-expressed in LUAD tumors. It is then plausible that SHP-2 activation may function to finetune PD-L1 expression levels. In immune cells, SHP-2 functions downstream of the PD-1:PD-L1 interaction by facilitating the internalization of the PD-1 receptor which ultimately results in the deactivation of the immune cell [20]. Likewise, it is conceivable that SHP-2 functions in a similar manner with regard to tumoral PD-L1 expression. The significance of SHP-2 co-expression with PD-L1 mRNA may be in a negative feedback loop, reducing PD-L1 levels once its expression is no longer necessary. Mutation of SHP-2 in malignant cells may alter SHP-2 activity or expression to disrupt this negative feedback loop, resulting in the aberrant constitutive expression of PD-L1 protein and continuous T-cell deactivation.

When we embarked on our studies, we most desired to understand how SHP-2 influences response to ICI therapy in KRAS-active tumors in order to direct our drug discovery efforts in a wet-lab setting. A limitation of the data deposited in the NCI-GDC is that the clinical data are often incomplete and lacking details on drug treatment and associated response or perhaps pre-date a particular therapy, like ICI in this case. However, we were able to address the expression of SHP-2 and PD-L1 in KRAS-active LUAD (~26% of the tumors). KRAS status did not change the outcome of the analysis. We identified other studies in which RNAseq data was collected from tumors treated with ICIs, one in melanoma and one in NSCLC [13, 14].

While the focus of our study is on NSCLC, treatment of melanoma using ICIs was approved a few years prior to use in NSCLC, and thus the data available in this cancer with respect to ICI treatment is more mature. It should be noted that melanomas rarely harbor KRAS mutations and more often HRAS mutations. Neither of the two small studies made the mutation status of Ras available.

The melanoma study was embarked by Regev and colleagues [13] and sought identify a gene expression profile that is associated with immune evasion that might predict response to ICI treatment. They conducted scRNAseq on melanoma tumors that were either untreated at the time of sequencing, or had acquired resistance to ICI therapy. These data allowed us to determine whether PTPN11 and CD274 gene expression associated with response to therapy. The authors of the study were more interested with defining signatures of resistance that could be used to screen patients prior to ICI therapy, so the experimental design was not ideal and the sample size was small. Importantly, PTPN11 mRNA levels were roughly equivalent between the treatment naïve and ICI resistant tumors. While this analysis provides some insight into the landscape of SHP-2 and PD-L1 coexpression, it is important to acknowledge that these tumors did not originate from the lung, have differing oncogenic mutations, and sample sizes were relatively low.

Finally, we used data from the NSCLC study carried out by Hwang and colleagues in which they sought to identify immune gene signatures that may predict clinical response to anti-PD-1 therapy [14]. The authors performed RNAseq on 21 NSCLC tumors that were divided by response to ICI therapy. For our analysis, we used average TPM values for PTPN11 and CD274 and compared tumors based on response to therapy. Expression of PTPN11 did not associate with DCB, but the tumors from patients who experienced DCB displayed increased expression of CD274 mRNA, consistent with other studies [21–23]. Taken together, these two studies do not suggest that the expression of PTPN11 mRNA is associated with to response to ICI therapy. Given these analyses considered alongside the TCPA analysis, it is likely that SHP-2 activity, not expression, bears more importance to PD-L1 expression, and subsequently response to ICI therapy, in NSCLC. Further, our findings suggest that reducing SHP-2 activity by pharmacological means would increase tumoral PD-L1 expression. Patients with PD-L1 expression >50% respond better to ICI therapy, supporting the potential for synergy of the co-inhibition of SHP-2 and PD-L1 in NSCLC [23–25].

This study outlines the significance of using of simple and efficient methods in real-world data analysis to further discovery efforts at the benchtop. Each study from which we gathered data had limitations that we have noted. The take-home message is that there is likely value in combining the use of molecules that inhibit the activity of SHP-2 and ICI in lung tumors and convince us that continued exploration into the role of SHP-2 on both PD-L1 expression is clinically important.

## Supporting information

**S1 Fig. SHP2_pY542 significantly correlates with phosphorylated proteins found in pathways that are SHP-2 targets. A**. Two-tailed non-parametric Spearman correlation analysis of RPPA protein expression data for Y542 phosphorylated SHP-2 and T202/Y204 phosphorylated MAPK **B**. Y527 phosphorylated Src kinase **C**. Y416 phosphorylated Src kinase **D**. Y705 phosphorylated STAT3 from 362 adenocarcinomas taken from The Cancer Proteome Atlas (TCPA: https://gdc.cancer.gov/about-data/publications/pancanatlas) LUAD-L4 dataset. The red line represents a linear regression line of best fit.
(TIF)

**S2 Fig. KRAS mutation status had no impact on PTPN11 and CD274 relationship.** Two-tailed non-parametric Spearman correlation analysis of bulk RNA-seq FPKM-UQ values taken from TCGA (GDC: https://gdc.cancer.gov/about-data/publications/pancanatlas) for 99 patients harboring mutations in the KRAS gene found in the TCGA-LUAD dataset. The red line represents a linear regression line of best fit.
(TIF)

**S3 Fig. Outlier analysis of PTPN11 TPM values for NSCLC response study.** Box and whisker plot of PTPN11 TPM values for NSCLC patients who did not respond to ICI therapy **(A)** or patients who did respond **(B)**. Outliers, highlighted in red, were determined by the 1.5 interquartile range (IQR) method which adds 1.5 times the IQR to the third quartile and excludes data points that fall above that value, and subtracts 1.5 times the IQR from the first quartile and excludes data points that fall below that value.
(TIF)

## Acknowledgments

We would like to thank the University of Kentucky College of Pharmacy for the support of this work, as well as John Gensel PhD, University of Kentucky, College of Medicine and Chi Wang, PhD, Markey Cancer Center, College of Public Health for consultation on statistical methods.

## Author Contributions

**Conceptualization:** Esther P. Black.

**Data curation:** Keller J. Toral, Mark A. Wuenschel.

**Formal analysis:** Keller J. Toral.

**Investigation:** Keller J. Toral.

**Methodology:** Keller J. Toral, Esther P. Black.

**Project administration:** Esther P. Black.

**Resources:** Esther P. Black.

**Software:** Keller J. Toral.

**Supervision:** Esther P. Black.

**Validation:** Keller J. Toral.

**Visualization:** Keller J. Toral, Esther P. Black.

**Writing – original draft:** Keller J. Toral, Esther P. Black.

**Writing – review & editing:** Keller J. Toral, Esther P. Black.

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
