## [Decision Letter · Decision Letter 0]

9 Jun 2021

PONE-D-20-25724

Real world genomic data supports combined use of SHP-2 and PD-1/PD-L1 inhibitors in solid tumors

PLOS ONE

Dear Dr. Black,

Thank you for submitting your manuscript to PLOS ONE. After careful consideration, we feel that it has merit but does not fully meet PLOS ONE’s publication criteria as it currently stands. Therefore, we invite you to submit a revised version of the manuscript that addresses the points raised during the review process.

We look forward to receiving your revised manuscript.

Kind regards,

Bing He

Academic Editor

PLOS ONE

Journal Requirements:

2. Please provide additional details regarding participant consent. In the ethics statement in the Methods and online submission information, please ensure that you have specified (1) whether consent was informed and (2) what type you obtained (for instance, written or verbal, and if verbal, how it was documented and witnessed). If your study included minors, state whether you obtained consent from parents or guardians. If the need for consent was waived by the ethics committee, please include this information.Once you have amended this/these statement(s) in the Methods section of the manuscript, please add the same text to the “Ethics Statement” field of the submission form (via “Edit Submission”).

Reviewers' comments:

Reviewer's Responses to Questions

**Comments to the Author**

1. Is the manuscript technically sound, and do the data support the conclusions?

Reviewer #1: No

Reviewer #2: Yes

2. Has the statistical analysis been performed appropriately and rigorously? 

Reviewer #1: Yes

Reviewer #2: Yes

3. Have the authors made all data underlying the findings in their manuscript fully available?

Reviewer #1: Yes

Reviewer #2: No

4. Is the manuscript presented in an intelligible fashion and written in standard English?

Reviewer #1: Yes

Reviewer #2: Yes

5. Review Comments to the Author

Reviewer #1: The manuscript, “Real world genomic data supports combined use of SHP-2 and PD-1/PD-L1 inhibitors in solid tumors” aims at establishing a relationship between SHP-2 and PD-1/PD-L1 using available patient genomic and proteome from data base.

The authors used statistical analysis to draw information from the existing data set and made a good attempt to test their hypothesis that inhibition of SHP-2 will improve the activity of ICI inhibitors that target PD-1 or PD-L1 in lung cancers. However, their inference of the data are extensions drawn based on inconclusive data.

In order to draw a correlation between SHP-2 and PD-L1, the authors analyzed TCPA data. As SHP-2 exists in active (phosphorylated) and inactive (unphosphorylated) form, it is important to analyze the levels of PD-L1 with reference to both SHP-2 forms. However, TCPA data set is missing a crucial control of inactive SHP-2. In the absence of this information, authors could have used another reference biomarker associated with SHP-2 signaling as an internal control. The figure 1 data does not establish a strong relationship between SHP-2 and PD-1 as the authors predicted/claimed.

The authors then analyzed data from melanoma tumors to support their hypothesis related to lung cancers. The two tumors possess their unique biomarkers and are different from each other in several ways to form comparable groups for analysis.

The data other than Figure 1A, establishes that there is no association between expression of PTPN11 and CD274 on the basis of mRNA levels. However, the authors extrapolated this to draw conclusion that it is SHP-2 activity and not the expression levels of SHP-2 that impacts PD-L1 expression. As stated before, in the absence of data for inactive SHP-2, there is no direct evidence to support this statement.

Additionally, sample size in Figure 2B and Figure 2C are very low to analyze the effect SHP-2 and PD-1/PD-L1 on ICI inhibitors.

Finally, there is no direct and clear evidence in the manuscript to support the title of this manuscript “Real world genomic data supports combined use of SHP-2 and PD-1/PD-L1 inhibitors in solid tumors”. The data from none of the figures or tables listed in the manuscript back this claim.

Reviewer #2: I read with interest this paper and given the limitations reported by the authors in terms of response data and trial results,

the strategy they followed is correct, but preliminary in impact.

With wider study assessment would become more robust, always as a preliminary study.

I wonder whether this weakness can be solved or otherwise (as per the current Title) the real-world data support of combined SHP-2 : PD-1/PD-L1 inhibitors use in solid tumors should be scaled a bit down.

6. PLOS authors have the option to publish the peer review history of their article (what does this mean?). If published, this will include your full peer review and any attached files.

Reviewer #1: No

Reviewer #2: No

---

## [Author Response · Author response to Decision Letter 0]

23 Jul 2021

Please refer to the submitted "Response to Reviewers" document.

---

## [Decision Letter · Decision Letter 1]

9 Aug 2021

Genomic data from NSCLC tumors reveals correlation between SHP-2 activity and PD-L1 expression and suggests synergy in combining SHP-2 and PD-1/PD-L1 inhibitors

PONE-D-20-25724R1

Dear Dr. Black,

We’re pleased to inform you that your manuscript has been judged scientifically suitable for publication and will be formally accepted for publication once it meets all outstanding technical requirements.

Kind regards,

Bing He

Academic Editor

PLOS ONE

Additional Editor Comments (optional):

Reviewers' comments:

Reviewer's Responses to Questions

**Comments to the Author**

1. If the authors have adequately addressed your comments raised in a previous round of review and you feel that this manuscript is now acceptable for publication, you may indicate that here to bypass the “Comments to the Author” section, enter your conflict of interest statement in the “Confidential to Editor” section, and submit your "Accept" recommendation.

Reviewer #1: All comments have been addressed

Reviewer #2: All comments have been addressed

2. Is the manuscript technically sound, and do the data support the conclusions?

Reviewer #1: (No Response)

Reviewer #2: Yes

3. Has the statistical analysis been performed appropriately and rigorously? 

Reviewer #1: (No Response)

Reviewer #2: Yes

4. Have the authors made all data underlying the findings in their manuscript fully available?

Reviewer #1: (No Response)

Reviewer #2: Yes

5. Is the manuscript presented in an intelligible fashion and written in standard English?

Reviewer #1: (No Response)

Reviewer #2: Yes

6. Review Comments to the Author

Reviewer #1: The authors have updated and modified the manuscript based of reviewers comments. This has considerably improved the paper.

Reviewer #2: (No Response)

7. PLOS authors have the option to publish the peer review history of their article (what does this mean?). If published, this will include your full peer review and any attached files.

Reviewer #1: No

Reviewer #2: No

---

## [Editor Report · Acceptance letter]

16 Aug 2021

PONE-D-20-25724R1 

Genomic data from NSCLC tumors reveals correlation between SHP-2 activity and PD-L1 expression and suggests synergy in combining SHP-2 and PD-1/PD-L1 inhibitors 

Dear Dr. Black:

I'm pleased to inform you that your manuscript has been deemed suitable for publication in PLOS ONE. Congratulations! Your manuscript is now with our production department. 

Kind regards, 

on behalf of

Dr. Bing He 

Academic Editor

PLOS ONE